# The Prevalence of Metabolic Syndrome and Its Components in Firefighters: A Systematic Review and Meta-Analysis

**DOI:** 10.3390/ijerph20196814

**Published:** 2023-09-23

**Authors:** Ashley Beckett, Jake Riley Scott, Angel Marie Chater, Louise Ferrandino, Jeffrey William Frederick Aldous

**Affiliations:** 1Institute for Sport and Physical Activity Research (ISPAR), University of Bedfordshire, Bedford MK41 9EA, UK; jake.scott1@study.beds.ac.uk (J.R.S.); a.chater@ucl.ac.uk (A.M.C.); louise.ferrandino@beds.ac.uk (L.F.); jeffrey.aldous@beds.ac.uk (J.W.F.A.); 2Centre for Behaviour Change (CBC), University College London, London WC1E 6BT, UK

**Keywords:** firefighters, metabolic syndrome, cardiovascular disease risk factors, obesity, hypertension, dyslipidemia, systematic review, meta-analysis

## Abstract

Previous studies consistently report a high prevalence of cardiovascular disease (CVD) risk factors among firefighters. However, the clustering of CVD risk factors, defined as metabolic syndrome (MetSyn), has received little attention by comparison. Therefore, the aim of this study was to estimate the pooled prevalence of MetSyn among firefighters. Using combinations of free text for ‘firefighter’ and ‘metabolic syndrome’, databases were searched for eligible studies. Meta-analyses calculated weighted pooled prevalence estimates with 95% confidence intervals (CI) for MetSyn, its components and overweight/obesity. Univariate meta-regression was performed to explore sources of heterogeneity. Of 1440 articles screened, 25 studies were included in the final analysis. The pooled prevalence of MetSyn in 31,309 firefighters was 22.3% (95% CI: 17.7–27.0%). The prevalences of MetSyn components were hypertension: 39.1%; abdominal obesity: 37.9%; hypertriglyceridemia: 30.2%; dyslipidemia: 30.1%; and hyperglycemia: 21.1%. Overweight and obesity prevalence rates in firefighters were 44.1% and 35.6%, respectively. Meta-regression revealed that decreased risk of bias (RoB) score and increased body mass index (BMI) were positively associated with an increase in MetSyn prevalence. Since one in five firefighters may meet the criteria for MetSyn, novel interventions should be explored to both prevent MetSyn and reduce the onset of CVD risk factors.

## 1. Introduction

Firefighters are required to be on-call twenty-four hours a day and are involved in rapid responses to emergencies, including fires, vehicle/machinery accidents and medical calls, among many more [1]. These often require the use of heavy (~25 kg) personal protective equipment, which can significantly increase cardiovascular strain [2]. Although improvements in technology have reduced the risk of fatalities during occupational tasks, mortality rates between 1990 and 2016 reveal that cardiovascular disease (CVD)-related fatalities are consistently the leading cause (~45%) of firefighter death [3].

The occupation of firefighting inherently promotes obesogenic behaviours, including increased food portion sizes, night-time snacking and sleep interruption [4]. Consequently, up to 50% of United States (US) firefighters showed significant (*p* < 0.001) increases in body mass index (BMI), total cholesterol and fasting blood glucose after a 5-year follow-up [5]. Overweight (25–29.9 kg·m^2^) and obesity (>30 kg·m^2^) have been associated with an increased prevalence of CVD risk factors, resulting in up to a three-fold risk of on-duty cardiac fatalities in firefighters [6]. However, BMI classification may over-represent firefighters categorised as overweight and obese, due to the increased muscle mass needed for occupational tasks [7]. For instance, the combined prevalence of overweight and obesity by BMI was ~80% in both US- and United Kingdom-based firefighters, but only between 43 and 63% when defined by body fat percentage (BF%), waist circumference (WC) or waist to hip ratio (WHtR) measurements [8,9].

Metabolic syndrome (MetSyn) is a condition of clustered CVD risk factors, which requires at least three out of the five components, including hyperglycaemia, hypertension, hypertriglyceridemia, dyslipidaemia and abdominal obesity [10]. The condition may have critical consequences for firefighters as those in the general public with MetSyn have a two-fold increase in age-adjusted CVD and a nearly three-fold increased risk for CVD-related deaths [11]. Furthermore, firefighters with even just one component of MetSyn are at a greater risk for an on-duty CVD-related fatality. For instance, firefighters with hypertension, hyperglycemia or dyslipidemia had a 4–12 times greater risk of on-duty death from CVD compared to their healthy counterparts [6,12].

With current data suggesting that firefighters’ obesogenic culture is escalating, the onset of CVD risk factors and MetSyn may play a significant role in the vast majority of future firefighter deaths [3]. To date, however, MetSyn among firefighters has been found to range widely across previous studies, with no established agreement on its prevalence. Therefore, the aim of this systematic review and meta-analysis is to provide a reliable estimate of the pooled prevalence of MetSyn and its components, as well as the factors associated with MetSyn among firefighters.

## 2. Materials and Methods

The protocol was registered with PROSPERO (registration number: CRD42021287974) and passed institutional ethical approval (2021ISPAR004). The processes follow the Preferred Reporting Items for Systematic Review and Meta-analysis (PRISMA-P) protocols [13].

### 2.1. Search Strategy

A systematic literature search was conducted on PubMed, Web of Science, EBSCO, ProQuest and Scopus for studies published from 1 January 1999 (the earliest validated definition of MetSyn) to 31 January 2023. The reference lists of all included studies were screened, and any unavailable full texts were requested from previous authors in the field. The search included multiple terms for population (“firefighter*”, “firemen”, “firewomen”, “fire service*”) and primary outcome themes (“metabolic syndrome*”, “Reaven syndrome”, “Syndrome X”, “Insulin resistance syndrome”, “Plurimetabolic syndrome”, “deadly quartet”, “cardiovascular disease risk”). All terms within each theme were combined with “OR”, and then the two themes were combined with “AND”.

### 2.2. Eligibility Criteria

After removing duplicate articles, titles and abstracts were screened independently by A.B. and J.S. The full text of potentially eligible articles was obtained to review eligibility for inclusion. The following criteria were used to select articles for inclusion in the review: (1) original studies from any country that were population-based, quantitative cross-sectional, longitudinal or prospective study design, published in the English language; (2) study participants were active whole time or retained fully qualified firefighters; (3) the study reports the prevalence of metabolic syndrome; and (4) the definition of metabolic syndrome was based upon validated criteria.

### 2.3. Data Extraction

Full texts of potentially relevant studies were assessed independently by two reviewers (A.B. and J.S.). All data from the included studies were extracted into Microsoft Excel using a piloted data extraction form. Data from each included paper were extracted by A.B. with a random 20% (assigned by a random number generator) reviewed independently by J.S. Information extraction from each article included the following items: publication data including author name, year of publication and title of manuscript. Methodological data including aims, target outcome, statistical analysis used, MetSyn diagnostic criteria, study design and inclusion/exclusion criteria. Participant data included sample size, mean age, sex, ethnicity, mass and height. Primary outcomes were extracted as raw cases of MetSyn and total number of participants. Secondary outcomes were extracted as raw cases of individual components of MetSyn (abdominal obesity, hypertension, dyslipidemia, hyperglycemia and hypertriglyceridemia), and cases of BMI were defined as healthy weight, overweight and obese, as defined by the criteria used in the respective study. Disagreements were resolved through discussions with J.A. Seven authors were contacted on two occasions requesting additional outcome data for the meta-analysis. Four of the seven authors replied with relevant outcome data. The three remaining papers were excluded due to irretrievable data.

### 2.4. Critical Appraisal of Included Studies

The risk of bias (RoB) for primary outcomes was assessed using the validated Joanna Briggs Institute (JBI) critical appraisal tool for prevalence studies [14]. The assessment was performed by A.B. on the target population, sample methods, sample size, statistical analysis methods, validity of measurement for the identification of MetSyn and response rate. Scoring of the JBI questions involved: “yes”, “no”, “unclear” or “not applicable”. The “yes” answers were used to determine the final score for each paper. Papers that reached <49% of questions scored as “yes” were classified as high RoB, 50–69% as moderate RoB and >70% as low RoB [15]. A second reviewer (J.S.) completed RoB for 20% of the chosen studies. Disagreements were resolved through discussions with J.A. Inter-rater reliability between the two reviewers (A.B. and J.S.) was strong (Cohen’s κ: 0.8) [16].

### 2.5. Data Analysis

Statistical analyses were conducted using R Studio version 4.0.4 (R Foundation for Statistical Computing). Packages “meta”, “metafor” and “weightr” were used to calculate weighted effect sizes for meta-analysis, perform meta-regression, generate forest plots and assess possible publication bias. Prevalence data were exported to R Studio in raw numbers (cases/total). Weighted average prevalence of MetSyn, components of MetSyn (hyperglycaemia, abdominal obesity, hypertension, hypertriglyceridemia and dyslipidemia) and BMI categories (healthy weight, overweight and obese) were then transformed using the Freeman–Tukey double arcsine transformation of proportions, which is the preferred method to stabilise variances. Random effect models were selected for all meta-analyses to allow for between-study variation reflecting sampling errors among other factors [17]. For the primary meta-analysis, a leave-one-out sensitivity analysis, whereby each paper was omitted in turn and the weighted prevalence re-calculated, was used to identify studies of disproportionate influence. The presence of possible publication bias in primary meta-analysis was assessed using the visual aid of a funnel plot and Egger’s test of funnel plot asymmetry [18]. Heterogeneity between studies was assessed using the *I*^2^ index. As a guide, *I*^2^ < 25% indicated low, 25–50% moderate and >50% high heterogeneity [19]. A univariate meta-regression was conducted to further explore potential sources of heterogeneity, including mean age, mean BMI, risk of bias score and year of publication.

## 3. Results

### 3.1. Study Characteristics

In total, 439 articles were identified after all duplicates were removed. After applying the exclusion criteria in the title and abstract screening, 397 articles were eliminated. Altogether, 54 articles were reviewed in full text, with 25 articles matching all inclusion criteria and, therefore, selected for the primary meta-analysis (Appendix A).

An overview of the study and participant characteristics are summarised in Appendix A. Within this systematic review of twenty-five studies, seventeen [20,21,22,23,24,25,26,27,28,29,30,31,32,33,34,35,36] were cross-sectional observational studies, six [24,26,37,38,39,40] were longitudinal and two [41,42] were prospective studies. All studies were published between the years 2009 and 2021. Studies varied by geographical location; nineteen of the studies were from the USA [20,21,22,23,25,27,28,29,31,34,35,36,37,39,41,42,43,44], two from Germany [24,38] and one each from Canada [40], South Korea [26], Iran [30] and Turkey [32].

Sixteen studies used the National Cholesterol Education Program Adult Treatment Panel III (NCEP/ATP III) [25,26,28,29,30,31,32,34,36,37,39,40,41,42,43,44], six used the Joint Interim Statement (JIS) [20,21,22,23,27,33], two used the International Diabetes Federation (IDF) criteria [24,38] and one used the American Heart Association and the National Heart, Lung and Blood Institute (AHA/NHLBI) definition to diagnose MetSyn. Definitions for components of MetSyn in all 25 included studies were as follows: hyperglycemia: fasting blood sugar: ≥100 mg/dL (5.6 mmol/L); hypertension: blood pressure ≥ 130/85 mmHg; hypertriglyceridemia: fasting triglycerides ≥ 150 mg/dL (1.7 mmol/L); and dyslipidemia: high-density lipoprotein cholesterol (HDL-C) male < 40 mg/dL (1 mmol/L) and female < 50 mg/dL (1.3 mmol/L). Minor differences occur when defining abdominal obesity where the JIS criteria are defined by population- and country-specific criteria; NCEP/ATP III and AHA/NHLBI criteria are WC ≥ 102 cm in males and ≥ 88 cm in females, and IDF requires WC ≥ 94 cm in males and ≥80 cm in females.

The BMI was reported in ten studies [20,23,24,27,29,30,32,34,37,38]. The mean BMI of the participants within the ten studies was 28.3 ± 3.5 kg·m^2^. The participants’ average age was 40.8 ± 8.2 y, with 98% being male. The median sample size was 447 (47 to 6947), and the sum of all participants from the 25 studies was 31,309. Total percentages from the eight studies [23,28,29,31,34,37,41,43] that reported ethnicity were 84% White, 6% Black, 4% Hispanic and 5% other.

The quality of all included studies is highlighted in Appendix A. Nineteen studies had low RoB scores [21,22,23,25,26,27,28,31,32,33,34,35,36,38,39,40,41,43,44], and six had moderate RoB scores [21,24,29,30,37,42]. The domain with the lowest RoB was the sampling of study participants, being low risk in twenty-four studies and with one unclear RoB [42]. The highest RoB domain was identified through sample size adequacy, being high risk in nine studies [24,27,29,30,32,35,36,37,38]. Furthermore, the requirement to ensure the measurement of MetSyn was conducted in a standardized and reliable manner was deemed unclear in fourteen studies [20,21,22,24,26,28,29,30,33,37,38,40,41,42].

### 3.2. Prevalence of Metabolic Syndrome

The MetSyn prevalence ranged from 8.9 to 56.9% across the twenty-five studies, with 31,309 firefighters included. The weighted average prevalence of MetSyn among firefighters was 22.3% (95% CI: 17.7–27.0%), with high heterogeneity for the random effects model (*I*^2^ = 97%, *p* < 0.001) (Figure 1).

### 3.3. Prevalence of Metabolic Syndrome Components

In individual random effects meta-analyses of MetSyn components, hypertension was the most prevalent component (39.2%), followed by abdominal obesity (37.9%), hypertriglyceridemia (30.2%), dyslipidemia (30.1%) and hyperglycemia (21.1%), respectively (Table 1). There was high heterogeneity for all MetSyn components (*I*^2^ = 97–98%, *p* < 0.001). Forest plots detailing the cases of MetSyn, the total numbers of firefighters within each study and the individual prevalence rates of each study can be found in Appendix A.

### 3.4. Prevalence of Overweight and Obesity

Individual meta-analyses showed that overweight was the most prevalent BMI category among firefighters at 44.1%. The average weighted prevalence of obesity was 35.6%, followed by a minority of firefighters in the healthy weight BMI category at 25.2% (Figure 2). For all BMI categories, there was high heterogeneity between studies (*I*^2^ = 92–98%, *p* < 0.001).

### 3.5. Meta-Regression

Univariate meta-regression revealed that MetSyn prevalence increased by 7% with every one unit decrease in RoB score (β = −0.07, SE = 0.01, *p* < 0.001) and increased by 5% with every one unit increase in mean BMI (β = 0.05, SE = 0.03, *p* = 0.037). Age (*p* = 0.403) and year of publication (*p* = 0.324) showed no statistically significant association with MetSyn prevalence (Figure 3).

### 3.6. Sensitivity Analysis

A leave-one-out sensitivity analysis revealed that two papers [35,37] may have exerted an unbalanced influence in the primary meta-analysis (Figure 4). Omitting [35] led to a small change in the meta-analytic estimate (19.2%). However, both studies also reported a high risk of bias for sample size criteria. A further analysis was therefore conducted without the inclusion of the two papers [35,37]. The revised weighted average prevalence estimate was 18.6% (95% CI: 14.7 to 20%), with slightly reduced but significant heterogeneity (*I*^2^ = 92%, *p* < 0.001). A funnel plot (Figure 4) indicated ambiguous publication bias; however, Egger’s linear regression test of funnel plot asymmetry informed no significant publication bias (*p* = 0.063).

## 4. Discussion

### 4.1. Summary of Evidence

This systematic review and meta-analysis examined the prevalence of MetSyn, components of MetSyn and overweight/obesity in firefighters across twenty-five studies. The main findings revealed that 22.3% of firefighters have ≥3 CVD risk factors and, therefore, meet the criteria for MetSyn. For MetSyn components, the highest prevalence rates were reported for hypertension (39.1%), followed by abdominal obesity (37.9%), hypertriglyceridemia (30.2%), dyslipidemia (30.2%) and hyperglycemia (21.1%), respectively. Secondary meta-analyses revealed around half of firefighters (44.5%) were overweight and just over one-third (35.6%) were obese when defined by BMI. A univariate meta-regression revealed that decreased RoB score and increased BMI were positively associated with an increase in MetSyn prevalence (Figure 3).

### 4.2. Metabolic Syndrome

This meta-analysis revealed that the pooled estimate of MetSyn prevalence among 31,309 global firefighters was 22.3% according to NCEP/ATP III, IDF, JIS and AHA/NHLBI definitions. This estimate is comparable with the estimated worldwide prevalence in the general population of 20–25% [45]. However, the prevalence of MetSyn was slightly lower than that found in other emergency worker populations. A meta-analysis of nine studies estimated the prevalence of MetSyn to be 26% in police officers according to NCEP/ATP III, IDF and AHA/NHLBI definitions [46]. Indeed, firefighters and police officers are exposed to somewhat comparable occupation stressors, including shift work, poor dietary habits and sleep deprivation that may increase the risk of developing MetSyn [47]. However, the observed decrease of MetSyn in firefighters may likely be a consequence of the ‘healthy worker effect’ from rigorous health screening in the recruitment process and continual annual fitness test requirements [48]. For instance, the National Fire Protection Association (NFPA) requirements suggest a firefighter must reach a minimum VO_2max_ of 42 mL·kg^−1^ as compared to 35 mL·kg^−1^ in police officers to maintain operational duty [49,50]. Further, there is promising evidence to suggest significant associations between increased cardiorespiratory fitness and reduced MetSyn outcomes in firefighters [20,43,44]. For instance, with every one-unit increase in VO_2max_ as assessed by metabolic equivalents, the odds of being diagnosed with MetSyn reduced by 31% in US firefighters [20]. It is likely, then, that current recruitment processes and annual fitness requirements may provide modest protective effects against the clustering of CVD risk factors and, therefore, reduce MetSyn prevalence compared to the general population and other emergency service workers.

### 4.3. Hypertension

Approximately 39% of firefighters had hypertension, and consequently, blood pressure greater than 130/85 mmHg was the most prevalent MetSyn component in the sub-group analysis. These findings are to be expected, given hypertension was also the most prevalent MetSyn component seen within the general population [45]. Specifically, global estimates report a prevalence of 40% hypertension, defined as blood pressure ≥ 130/85 mmHg [45]. Hypertension is well established as the leading preventable risk factor for CVD and all-cause mortality worldwide [51]. The cause of the observed high prevalence of hypertension and abdominal obesity was not specifically investigated in this study. Nonetheless, common risk factors include obesity, alcohol consumption, physical inactivity and unhealthy diet, which have all been shown to be highly prevalent issues among firefighters [47,52,53]. Furthermore, there is a general consensus that occupational risk factors, including noise exposure from alarms/sirens, post-traumatic stress disorder and shift work, likely contribute to the high prevalence of hypertension in firefighters [47]. Studies have also reported that a low proportion of hypertensive firefighters receive treatment [12,54,55]. Firefighters were found to have the lowest awareness (51% aware), treatment (79% treated) and prevalence of controlled blood pressure (48% controlled) across 13 different occupational groups [54]. This is concerning as firefighters with hypertension are seen to be at a 3.5 times greater risk for sudden cardiac death than those with healthy blood pressure, and furthermore, healthy firefighters were ~4 times (OR: 3.77) more likely to survive a CVD-related event than their hypertensive colleagues [55]. Consequently, despite recommended cardiorespiratory fitness standards, firefighters are inherently exposed to a variety of occupational stressors likely to increase hypertension, even without MetSyn diagnosis.

### 4.4. Abdominal Obesity

Abdominal obesity was present in 37.9% of firefighters and was the second most prevalent MetSyn component. These findings reflect a ~3% increase in prevalence compared to the general population (34.9%) when defined by WC ≥ 102 cm for men or ≥88 cm for women [45]. Furthermore, it also reveals a considerable reduction (~15%) in elevated WC compared to police officers at 52.6% [56]. Interestingly, there was a ~2% increase in abdominal obesity prevalence in comparison to defining obesity by BMI in the present study. This extends previous research, which documented that obesity prevalence was, in fact, highest by WC (24.2%), followed by BMI (23.1%) and BF% (17.3%) in US firefighters [8]. Furthermore, both Russian and Brazilian firefighters had a higher prevalence of elevated WC (28% and 19%) compared to BMI obesity (22% and 11%), respectively [57,58]. The causes for such a high prevalence of abdominal obesity in firefighters are likely similar to those mentioned for hypertension. However, the implications of the condition differ as elevated WC is a direct measure of visceral adiposity, which has been suggested to be a causal factor in increasing the risk of insulin resistance, type 2 diabetes and CVD [59]. Thus, a high proportion of firefighters with abdominal obesity may go on to develop a clustering of other CVD risk factors and, subsequently, MetSyn through a variety of pathophysiological mechanisms [59].

### 4.5. BMI-Defined Overweight and Obesity

Secondary meta-analyses revealed the pooled prevalence of BMI-defined overweight and obesity among firefighters was 44.1% and 35.6%, respectively. These data are considerably higher than global (39% and 13%), US general public (39 and 32%) and other emergency service worker (43% and 24%) estimates for overweight and obesity, respectively [60,61,62]. The meta-regression conducted in the present study also found a significant positive association between increased BMI and increased MetSyn prevalence. Specifically, for every 1-unit increase in mean BMI, the prevalence of MetSyn was estimated to increase by ~5%. These data are consistent with previous research identifying that healthy-weight firefighters had a 2% prevalence of MetSyn, which increased to 6% in overweight and 18% in obese firefighters [31]. Furthermore, firefighters with BMI-defined overweight and obesity were 3 and 9 times more likely to have MetSyn compared to their healthy counterparts, independent of age [31]. Obesity has been long established as an independent risk factor for CVD and is known to be linked to risk factor clustering, likely due to the buildup of excess adiposity augmenting several pro-inflammatory states, including decreased insulin sensitivity and adipocyte tissue dysregulation [59]. Despite this, BMI has also been shown to overestimate overweight and obesity prevalence compared to BF% or WC in firefighters, likely due to the inability to differentiate between muscle mass and fat mass [8,9]. It seems, however, that this is primarily a consequence of overestimating BMI defined as overweight (i.e., 25–29.9 kg·m^2^) rather than obesity (≥30 kg·m^2^) [8]. For instance, the prevalence of overweight by BMI (58.1%) was considerably greater than when assessed by BF% (43.5%) or WC (27.6%) in UK firefighters [9]. However, the prevalence of obesity showed much greater agreement at 22.3%, 19.1% and 15.5% for BMI, BF% and WC, respectively [9]. Thus, given the practicality of BMI to be used in occupational health settings, this measure may continue to provide a resourceful measure of overall CVD health in firefighters. Furthermore, the high prevalence of obesity defined by both elevated WC and BMI within the present study calls for substantial efforts to be made to both prevent and reduce these conditions in firefighters for the benefit of occupational health, safety and performance.

### 4.6. Hyperglycemia, Dyslipidemia and Hypertriglyceridemia

The prevalence of hyperglycemia (21.1%) and hypertriglyceridemia (30.2%) in firefighters were broadly similar to global estimates at 24.5% and 28.9%, respectively [45]. Whereas dyslipidemia (30.1%) was ~10% lower in firefighters than in the general population (40.2%) [45]. The observed decrease in dyslipidemia may likely be a consequence of one study (with a moderate RoB at 56%) in 97 German firefighters showing a 3% prevalence of dyslipidemia [24], which was substantially (15–59%) lower than the other 10 studies included in the meta-analysis. In general, 77% of the studies included in this meta-analysis were from the US. Considering 65% of US firefighters are volunteers and may work second jobs while being on-call for emergency deployment [63], it is unsurprising that firefighters’ health somewhat reflects that of the general public. Furthermore, the mean age for all firefighters included in this meta-analysis was 40.8 ± 8.2 y, representing an age group (in both general population and firefighters) with increased odds of developing CVD risk factors [34]. Therefore, it is likely that current firefighter fitness requirements alone may not sufficiently prevent the onset of CVD risk factors in firefighters, analogous to the general public.

### 4.7. Strengths and Limitations

The quality of evidence for the prevalence of MetSyn was, on average, high, and the majority (77%) of RoB ratings were judged as low. Ensuring MetSyn was measured in a standard, reliable way for all participants was the most unclear parameter (64%), revealing issues with inadequate reporting details essential for judgments of study quality. The highest RoB came from the adequacy of the sample size (27%), and none of the included studies provided a sample size justification. Using recommendations from the JBI critical appraisal checklist, a sample size calculation for future prevalence studies reporting MetSyn should exceed 246 participants to estimate population prevalence with adequate precision [64]. Therefore, nine [24,29,30,32,35,36,37,38,43] of the twenty-five studies included in this meta-analysis did not reach this sample size threshold, potentially leading to less precise confidence intervals and overall prevalence estimates [64]. Although Egger’s test confirmed no publication bias (*p* = 0.063) in the primary meta-analysis, an overestimation in MetSyn prevalence in studies with smaller sample sizes was evident in the Baujat plot (Figure 4).

Meta-regression analysis confirmed that reduced RoB score was significantly associated with an increase in MetSyn prevalence. Specifically, MetSyn prevalence increased by ~7% with every one-unit decrease in RoB score. On further investigation to explore these results, only 36% of studies stated the specific fasting duration prior to data collection, despite fasted blood measurements as essential criteria for MetSyn diagnosis. This is concerning given that fasting duration can cause significant inaccuracies in all included blood markers, i.e., blood glucose, triglycerides and HDL-C [10]. These inaccuracies are also likely to be exacerbated in post-prandial states, which are characterised by increased glucose, insulin and triglyceride levels, which may overestimate the prevalence of CVD risk factors [65]. Although conducting health research on firefighters may provide practical limitations due to their spontaneous work/shift schedules, future studies should provide adequate detail in fasting duration prior to blood collection for the diagnosis of MetSyn.

The present study has some limitations, most notably the lack of subgroup analysis to further explore MetSyn prevalence and the potential influencing factors in firefighters. Although sub-group analysis for diagnosis tool, gender and location were planned, after extracting the data, it was revealed that there were vast disparities in the number of studies per sub-group. For diagnostic criteria, sixteen of the twenty-five studies used NCEP/ATP III, with only a minority using other definitions such as JIS (6), IDF (2) and AHA/NHLBI (1). Between these definitions, only modest differences occur when defining abdominal obesity [10]. Thus, there is relative homogeneity when defining MetSyn within this meta-analysis. Nonetheless, the present finding suggests MetSyn prevalence may primarily be generalised according to the NCEP/ATP III criteria.

Nineteen of the twenty-five studies were from the US, with 98% of the participants being male. It could be hypothesised that variations in work schedules, cultural differences in dietary/physical activity behaviours and healthcare systems may significantly impact the prevalence of MetSyn and its components among firefighters from different countries. Relatedly, the US general population currently has the highest prevalence of MetSyn at 46.4%, considerably higher than European (26.8%), Asian (28.5%) and Mediterranean regions (35.9%) [45]. Thus, it could be suggested the present findings in this meta-analysis may overestimate the prevalence of MetSyn in firefighters on a global/regional scale. Therefore, future research should make efforts to explore these data among a variety of different countries. Additionally, although gender’s influence on MetSyn outcome is largely inconclusive, in females, dyslipidemia, abdominal obesity and hyperglycemia have been shown to be significantly larger contributors to the prevalence of MetSyn compared to males [66]. With the increasing number of female firefighters within the US Fire Service [67], it is vital to further assess MetSyn in females as a paucity of research currently exists.

Finally, as is common in meta-analysis of prevalence data, there was substantial heterogeneity across all meta-analyses [68]. The rigorous selection criteria ensured that all the studies included in the analysis shared enough similarities, allowing them to be pooled effectively. Further, the use of random-effect meta-analysis models within the present study specifically considers the between-study heterogeneity and the within-study sampling error [68]. Nonetheless, the present findings should be interpreted with careful judgement and considered as estimates only.

## 5. Conclusions

This is the first systematic review and meta-analysis to estimate the pooled prevalence of MetSyn and its components among firefighters worldwide. The present findings indicate a high prevalence of MetSyn within such a physically demanding and high-stress occupation. Further, a high prevalence of hypertension and abdominal obesity were also observed. It is necessary that these data are further investigated in a wider demographic of firefighters from a larger sample of countries. Novel interventions tailored for firefighting should be developed to combat the high prevalence of CVD risks in this population.

### Protocol Registration

Details of the protocol for this systematic review were registered on PROSPERO (CRD42021287874) and can be accessed at https://www.crd.york.ac.uk/prospero/display_record.php?ID=CRD42021287974 (accessed on 6 September 2023).

## Figures and Tables

**Figure 1 ijerph-20-06814-f001:**
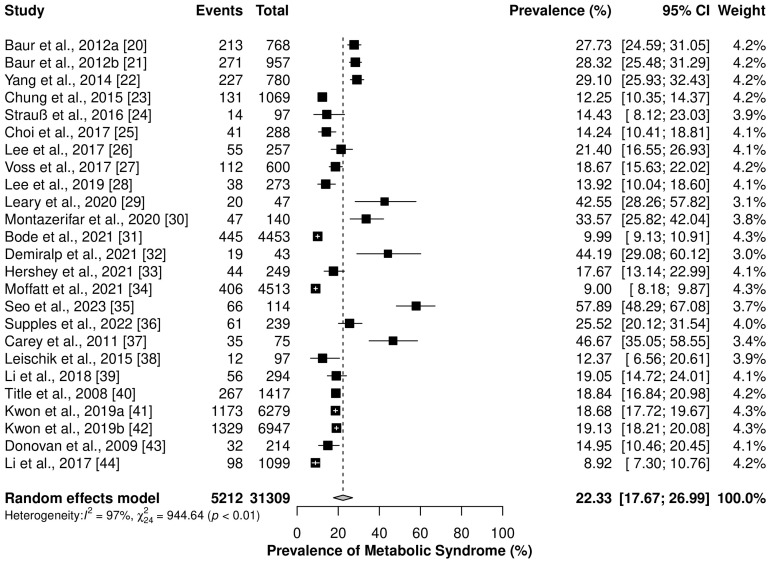
Forest plot for pooled prevalence of metabolic syndrome for all included studies. Events = number of firefighters with metabolic syndrome; Study = Author + Year + Reference number; Total = total number of firefighters in the study. Black squares represent the prevalence effect size. White crosses/black horizontal lines represent 95% CI [20,21,22,23,24,25,26,27,28,29,30,31,32,33,34,35,36,37,38,39,40,41,42,43,44].

**Figure 2 ijerph-20-06814-f002:**
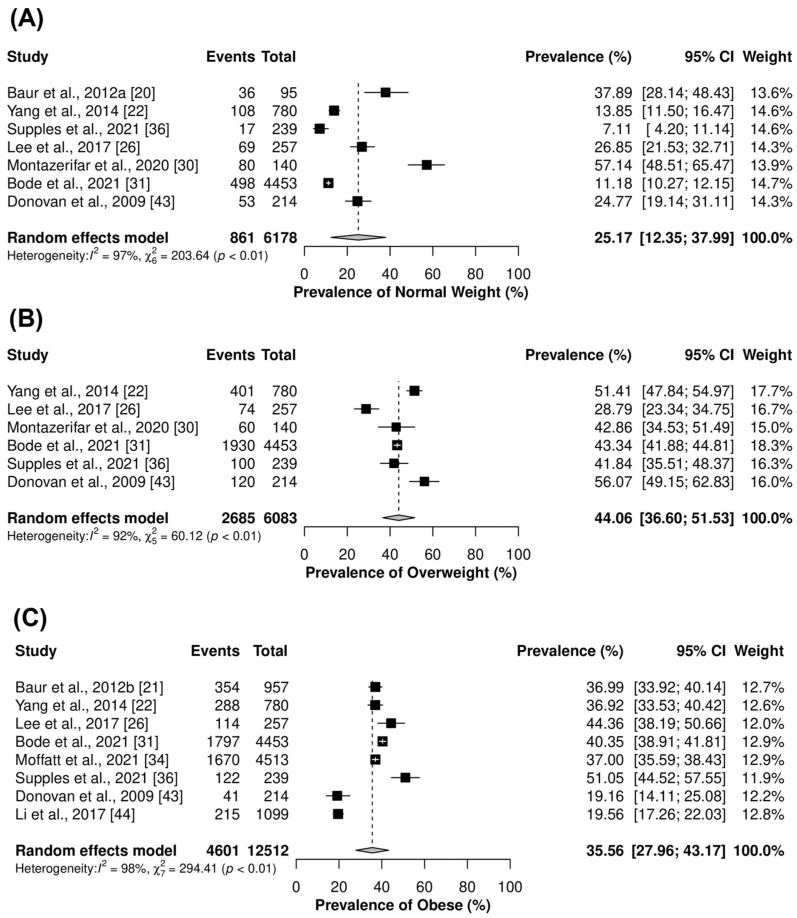
Forest plots of prevalence of (**A**) Healthy weight, (**B**) Overweight and (**C**) Obese, respectively. BMI defined as <24.9 kg·m^2^ = Healthy weight; 25.0–29.9 kg·m^2^ = Overweight; >30 kg·m^2^ = Obese. Events = number of firefighters within each respective BMI category; Study = Author + Year + Reference number; Total = total number of firefighters in the study. Black squares represent the prevalence effect size. White crosses/black horizontal lines represent 95% CI [20,21,22,26,30,31,34,36,41,44].

**Figure 3 ijerph-20-06814-f003:**
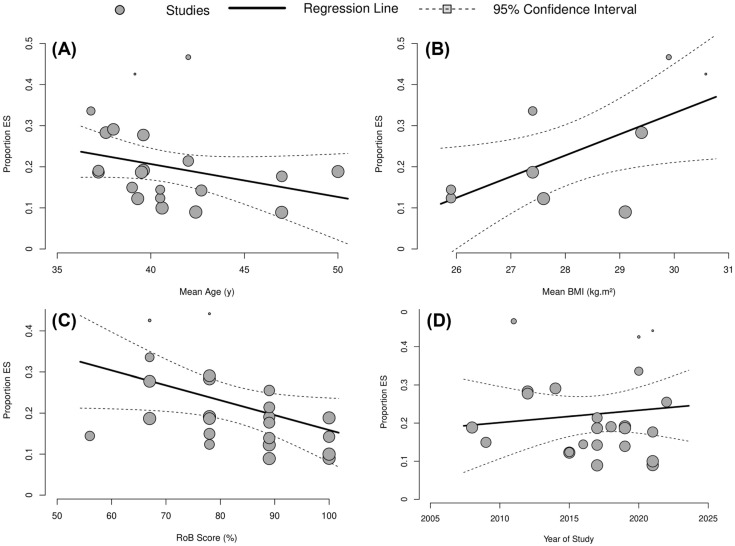
Meta-regression bubble plot to explore associations between (**A**) Mean Age, (**B**) Mean BMI, (**C**) RoB Score and (**D**) Year of Study on the prevalence of MetSyn.

**Figure 4 ijerph-20-06814-f004:**
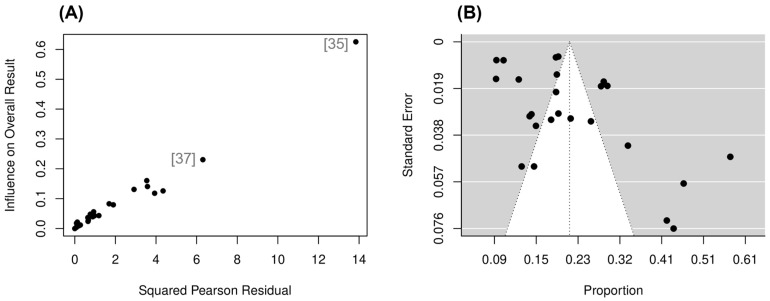
(**A**) Funnel plot of included studies in primary meta-analysis [35,37]. (**B**) Baujat plot for all included studies. Grey numbers in brackets represent individual studies included in meta-analysis (correspondence between numbers and studies can be found in reference list).

**Table 1 ijerph-20-06814-t001:** Pooled prevalence of MetSyn components in reported studies.

Component	*n* Studies	*n* Cases	*n* Total	Pooled Prevalence * (95% CI)	Min, %	Max, %	*I*^2^, %	*p* Heterogeneity
Hyperglycemia	11	579	6392	21.12 (11.63–30.61)	0.93	52.0	98	<0.001
Abdominal Obesity	11	2305	6392	37.93 (26.39–49.46)	19.16	87.14	98	<0.001
Hypertension	11	1006	2836	39.19 (28.75–49.62)	11.24	68.09	97	<0.001
Hypertriglyceridemia	11	1100	6392	30.23 (22.33–38.13)	12.01	61.43	97	<0.001
Dyslipidemia	12	2041	7349	30.12 (21.16–39.07)	3.09	62.14	97	<0.001

Min = Minimum reported prevalence; Max = Maximum reported prevalence; * Random effects model.

## Data Availability

All data generated or analysed during this study are included in the published review article.

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
