# Peer review of "The Prevalence of Metabolic Syndrome and Its Components in Firefighters: A Systematic Review and Meta-Analysis"

_ijerph, 2023, doi:10.3390/ijerph20196814_

Round 1

Reviewer 1 Report

The manuscript is a meta-analysis of prevalence of metabolic syndrome and its components in firefighters. More than 1,400 articles have been screened. The final analysis included 25 studies, with more than half from US. The result is interesting and important to study firefighter's health. There are several concerns listed below.

1. The number of total articles in abstract is not consistent with that in the PRISMA diagram.

2. What statistical method/model was used for the result presented in Figure 2? How the subgroup prevalence was combined to calculate the overall average prevalence, when there are overlapped and different studies among subgroups?

3. How was the weight determined in Figure 1 and Figure 2?

4. More variables should be investigated in the meta-regression analysis, such as diagnosis tool, gender, and location/race group.

5. In paragraph 4.6, does -4% and +2% indicate difference comparing to global estimates? What are the global estimates?

6. The heterogeneity is high for all analyses. The result may not be meaningful and generalizable to entire firefighters group.

NA

Author Response

Dear Reviewer’s and Journal Editor,

The authors thank the Reviewers for their advice and insight.

Please find attached a point-by-point rebuttal or note of changes in-line with specific and general Reviewer comments. Within the manuscript such changes have been implemented and highlighted using red font, and these changes follow the original ‘Word’ line numbering system i.e. 1 – 607 Extracts of revised/newly included text from the manuscript are included below to supplement rebuttal comments, such extracts are identifiable from being presented in italics. We hope the Reviewer’s are satisfied with our revisions and agree that the manuscript has greatly benefitted from this process.

Reviewer 2 Report

I have reviewed the manuscript entitled The prevalence of metabolic syndrome and its components in 2 firefighters: A systematic review and meta-analysis. It is an interesting and clear manuscript that deserves to be published.

My comments are as follows:

1.- Line 123: “…transformation of proportions; the preferred method to stabilise variances.” Clarify the sentence (are some words missing? Should “;” be deleted?).

2.- “The overall weighted average prevalence of having at least 1 component of MetSyn was 31.7%”. Please clarify how did you obtain information of firefighters having 1 component of the MetSyn, since the manuscripts reviewed were about firefighters having MetSyn, which implies having at least 3 components.

Author Response

(The authors gave the same response as above.)

Reviewer 3 Report

The Authors of the current systematic review and meta-analysis aimed to provide a reliable estimate of the pooled prevalence of metabolic syndrome (MetSyn) and its components as well as the factors associated with MetSyn among firefighters.

The manuscript is clearly written. The research design is appropriate and the methods are adequately described. The results justify the conclusions. However, some minor remarks are as follows:

-        The aim of the study in the Abstract section should be clearly stated.

-        Some abbreviations in the Abstract section need an explanation.

-        The conclusion section should be re-written, i.e. without repeating the results.

-        A subtitle „6. Patents” should be deleted.

Author Response

Dear Reviewer’s and Journal Editor,

The authors thank the Reviewers for their advice and insight.

Please find attached a point-by-point rebuttal or note of changes in-line with specific and general Reviewer comments. Within the manuscript such changes have been implemented and highlighted using red font, and these changes follow the original ‘Word’ line numbering system i.e. 1 – 607 Extracts of revised/newly included text from the manuscript are included below to supplement rebuttal comments. We hope the Reviewer’s are satisfied with our revisions and agree that the manuscript has greatly benefitted from this process.
